# Impact of the surplus distribution principle on the development of agricultural cooperatives in China

Yugang Han[1,2,☯,*], Yating Nie[1,‡], Aihua Lv[1,‡], Lei Ye[1,2,☯,‡]

1 Economic Management Institute of Anhui Normal University; Wuhu, Anhui, China, 2 Rural Revitalization Research Institute of Anhui Normal University, Wuhu, Anhui, China

☯ These authors contributed equally to this work.
‡ YN, AL and LY also contributed equally to this work.
* hyg1005@ahnu.edu.cn

## Abstract

Surplus distribution is at the core of cooperative systems. This study examines the impact of the surplus distribution principle on the development of agricultural cooperatives. We utilize panel data on cooperative development and surplus distribution for 30 provinces in China from 2008 to 2021 to identify the development pattern and surplus distribution status of cooperatives. A two-way fixed effects model is used as a benchmark to examine the impact of the surplus distribution principle on the development of cooperatives. The empirical analysis shows that the surplus distribution method, which strictly adheres to a trading return of more than 60% of the distributable surplus, has a significant negative impact on cooperative membership size but a significant positive impact on cooperative income. These results remain robust after controlling for the estimation method, replacing the core explanatory variables, shrinking the tails, and conducting endogeneity tests. This study contributes to a macro-level understanding of the impact of the surplus distribution principle on the performance and scale of cooperatives and provides important policy insights for promoting surplus distribution standardization in farmers' cooperatives and guiding members to establish a stable benefit-linkage mechanism within cooperatives.

## 1 Introduction

Cooperatives have emerged as a critical organizational form in modern agriculture, particularly in developing countries where smallholder farmers face significant challenges in accessing markets, achieving economies of scale, and coping with market risks. A cooperative is an association of self-governing farmers who voluntarily unite to meet common social, economic, and cultural needs through collective ownership and democratic control. Globally, cooperatives have contributed to long-term economic growth by providing quality employment to 280 million people and facilitating

**Data availability statement:** All relevant data are within the paper and its Supporting Information files.

**Funding:** Funded by: Anhui Normal University Talent Cultivation Fund Project "Research on the Activation of Rural Collective Economy Development by New Cooperatives Led by Party Organizations" (QZJDBN2021XZC04); Anhui Provincial Higher Education Research Project (Major Project) "Research on the Mechanisms and Effects of Collaborative Development between Anhui's Wanjiang and Northern Anhui Regions—Based on a 'Subject-Industry-Space' Collaborative Perspective" (2023AH040017).

**Competing interests:** The authors have declared that no competing interests exist.

social governance, technology adoption, and risk management [1–3]. Furthermore, cooperatives help reduce transaction costs, mitigate market risks [4], and promote economies of scale [5]. Thus, cooperatives play an increasingly important role in agriculture in developing countries [6,7].

Compared to developed countries, cooperatives in developing countries tend to be small, with fewer members, lower levels of mechanization and specialization, and are mostly village-based [8]. However, as a new organizational form to promote agricultural modernization [9], agricultural cooperatives play an increasingly important role in agriculture in developing countries [6,7]. As political, economic, and sociocultural conditions vary among countries, the development trajectories of cooperatives are quite different, and cooperative development models and distribution patterns have shown significant heterogeneity.

Kumar et al. [9] modeled the membership of local dairy cooperatives in India to demonstrate that the benefits of cooperatives have a greater impact on small farmers. Montero [10] showed that cooperatives in El Salvador are biased toward growing food crops over haciendas. This is because cooperative members have the option to consume some of their staple crops, creating greater incentives to participate. Vietnam's Cooperative Law stipulates that surplus income must be distributed primarily according to the proportion of members' use of products and services, as well as the labor members' contribution to the job-creating cooperative, with the remainder distributed according to the amount of capital contributed. Furthermore, Vietnam's Law on Cooperatives provides for the division of responsibilities and obligations according to the amount of capital contributed, meaning that members who contribute more have greater obligations. However, Trach [11] has argued that Vietnamese cooperations distributing surpluses primarily to members who use the products and services is unreasonable and does not guarantee contributor ownership.

China's first Cooperative Law was enacted in 2007, marking a significant milestone in formalizing cooperative principles. This legislation mandates that cooperatives prioritize farmer membership, voluntary participation, democratic management, and surplus distribution based on transaction volume. The surplus (revenue minus costs) of farmers' professional cooperatives differs from the profits of enterprises because the goal of cooperatives is to serve their members and ensure that they receive the best service at the lowest cost rather than to maximize surplus. China's Law on Farmers' Specialized Cooperatives stipulates that the current year's surplus is the distributable surplus of farmers' specialized cooperatives after compensating for losses and withdrawing the provident funds.

Agricultural cooperatives have become pivotal to agricultural modernization in China, which is the largest developing country worldwide and has a substantial rural population. By September 2023, the number of agricultural cooperatives in China exceeded 2.22 million, covering nearly half of the country's agricultural households. Despite this impressive growth, Chinese cooperatives face persistent challenges, such as low overall development levels and uneven regional progress [6,7,12]. These bottlenecks highlight the need for more effective governance mechanisms and strategic policy interventions.

Surplus distribution is central to the institutional construction of farmers' professional cooperatives and crucial for maintaining member relationships and organizational stability [13]. Academic discourse on surplus distribution in cooperatives has presented divergent perspectives. Some scholars have argued that strict adherence to standardized distribution principles may hinder performance because cooperatives will struggle to balance standardization with financial outcomes [14,15]. In contrast, others have posited that governance rules and central policies can enhance accountability [16–18]. The lack of a regulated surplus distribution system can undermine cooperative performance, leading to elite capture and the potential dissolution of cooperatives [19]. Thus, an elite orientation undermines the longevity and sustainability of cooperatives [20,21].

However, empirical research on the relationship between surplus distribution principles and cooperative development remains limited, especially in the context of Chinese agricultural cooperatives. This study aims to narrow these gaps by investigating the impact of the surplus distribution principle stipulated in China's Cooperative Law on agricultural cooperative development. Specifically, we focus on two key dimensions: the size and income of cooperatives. Using panel data from 30 provinces in China over 14 years (2008–2021), we employ a two-way fixed effects model to assess the effects of surplus distribution based on transaction returns.

This study makes three main contributions to the literature. First, we extend the research on new institutional economics by examining the role of surplus distribution in the development of farmers' cooperatives, an organizational form that has received limited attention in this context. Second, our empirical analysis provides robust evidence of the effects of surplus distribution on both cooperative size and income, offering new insights into the trade-offs involved in adhering to standardized distribution principles. Third, the findings of this study are of practical significance to policymakers and cooperative leaders in developing countries and provide important policy insights for promoting the standardization of surplus distribution in developing countries and guiding members to establish a stable mechanism for linking interests with cooperatives.

The remainder of this paper is structured as follows. Section 2 presents the theoretical framework and research hypotheses. Section 3 describes the data, model, and variables used in the analysis. Section 4 provides an overview of Chinese cooperatives' development trends and surplus distribution practices. Section 5 presents the empirical results. Finally, Section 6 concludes the paper with policy implications and suggestions for future research.

## 2 Theoretical analysis and research hypotheses

### 2.1 Theoretical analysis

Coase's "The Nature of the Firm," published in 1937, is regarded as the founding work of new institutional economics, which has developed four basic theories: transaction cost theory, property rights theory, firm theory, and institutional change theory. Furthermore, the analysis scope has been extended to almost all social science fields. Property rights are the core content and main research object of new institutional economics. New institutional economists generally believe that property rights are the behavioral relationships between people recognized by the existence of things and their use [22]. First, property rights are an exclusive combination of rights including exclusive ownership, exclusive use, income exclusivity, and free transfer. Second, property rights are a set of legal constraints based on a contractual relationship between people as a result of their relationship with things. Third, property law defines people's expected benefits, places strict limits on the boundaries of individual responsibilities and obligations in exchange, and governs people's behavior toward each other. Coase's third theorem states that the system for the supply of property rights is the basis on which people conduct transactions and optimize resource allocation. Through a unique institutional design, property rights can promote economic development, which is the basic pathway through which property rights generate economic benefits.

Cooperatives are member-owned; therefore, their governance structures and mechanisms may differ from those of other types of investor-owned organizations [23]. The type of distribution system adopted by the cooperative is related to the residual control versus residual claim rights of the cooperative [24]. The surplus distribution principle is the refraction

and embodiment of cooperatives' property rights and governance structures and the essential difference between cooperatives and corporations. According to the theory of property rights, the ownership and income rights of cooperatives belong to all members, and this interest is exclusive. From the perspective of new institutional economics, surplus distribution is considered a property right in which all community members enjoy ownership and residual claim rights. The surplus distribution system owned by the members of a cooperative society allocates residual claim rights to each member and provides a property right basis for members to participate in surplus distribution. The practical necessity of surplus distribution also lies in improving member cohesion, protecting the interests of small and weak members, encouraging members to pay attention to the cooperative's sustainable operation, and forming a benefit- and risk-sharing mechanism. In a cooperative property rights system, granting ownership to members may have an incentive effect. However, the distribution of benefits and surpluses among members may also lead to free-rider problems, which can negate the incentive effects of cooperative ownership [10].

Regarding the specific distribution method for cooperative surpluses, since the establishment of the Rovindale principle, the secondary rebate has become a classic principle of cooperatives. This means that when members trade their products, in addition to receiving part of the profits (primary concessions), they also receive a share of the final surplus based on transaction volume (amount). The Law of the People's Republic of China on Farmers' Cooperatives also stipulates that "the surplus shall be returned mainly in proportion to the volume (amount) of transactions between the members and the farmers' cooperative, and the total amount of return shall not be less than 60 percent of the distributable surplus." This type of distribution method has a deep practical foundation and meaning, and most cooperative surpluses are generated from transactions between members and cooperatives.

## 2.2 Research hypotheses

### 2.2.1 Impact of the surplus distribution principle on cooperative size.
In a collective, if the total number of members increases excessively, each member receives less of the common good. As the cooperative continues to expand, an otherwise loose organizational structure reduces the likelihood of direct supervision among members. This lack of supervision makes it difficult for managers to make accurate and timely determinations of whether members consciously fulfil their obligations and comply with the rules. This objectively indulges cooperative members who wish to pursue benefits alone without being willing to pay for them accordingly, thus contributing to the phenomenon of free-riding members, which may result in a collective organizational dilemma [25].

Therefore, when members receive the same benefits as others, despite not paying the same amount to the cooperative, the motivation of those who pay more will decrease, which is inconducive to scale growth [26]. Furthermore, compared to large producers, small farmers trade small amounts of agricultural products and generate lower benefits for cooperatives. According to the theory of new institutional economics, the essence of surplus distribution is a type of property right, and all members are the subject of property rights, enjoying the rights of ownership and income. Therefore, when heterogeneity exists among cooperative members, the principle of surplus distribution, which is mainly based on the return on transaction volume (amount), affects the incentives for relatively capital-rich farmers to join cooperatives.

Furthermore, cooperatives should have a moderate membership size, as excessive scale may affect members' commitment to the cooperative. Increasing membership size reduces average costs and improves the bargaining power of cooperatives in trading with other market entities. However, a large member presence may result in excessive organizational costs, leading to collective action dilemmas. Expanding membership size may lead to an oversupply of products and difficulties for cooperatives in marketing their products, thus undermining their overall production efficiency [27]. Liang [28] found that the relationship between size and performance in cooperatives follows an inverted "U" shape, as shown in **Fig 1**.

A surplus distribution system based on the return of transactions requires a linkage mechanism of "benefit and risk sharing" to be established, thus limiting the number of "free-rider" members to a certain extent and controlling the size of cooperatives within a reasonable range. Therefore, we propose the following:

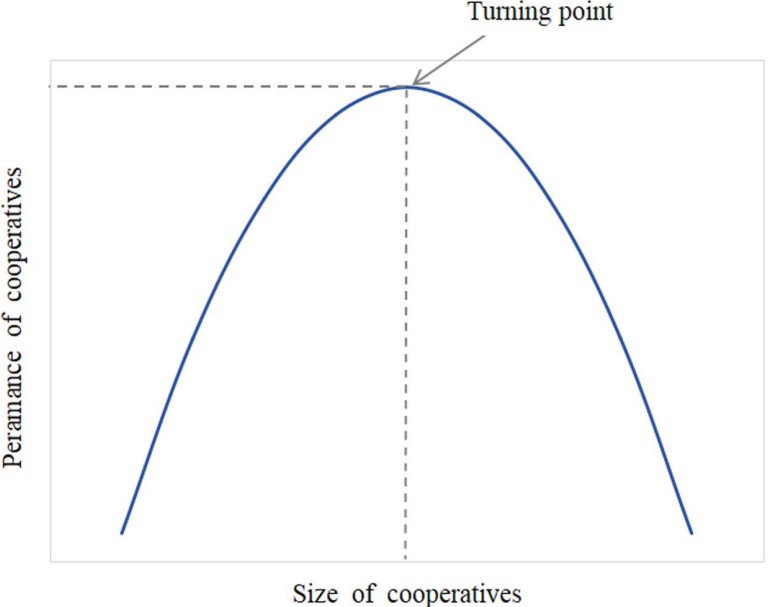

**Fig 1. Inverted U-shaped relationship between cooperative size and performance.**

Hypothesis 1: The principle of surplus distribution on a trade basis has a dampening effect on the size of cooperatives compared to cooperatives that do not return on a trade basis.

**2.2.2  Impact of the surplus distribution principle on cooperative income.**  Cooperatives should establish a rational distribution system, as an irrational distribution system often leads to "free-riding" and slack work, which are inconducive to their healthy development. If the surplus distribution of cooperatives considers the residual claims of different types of members, it can help alleviate conflicts of interest within cooperatives and increase the enthusiasm of members with different resource endowments to join cooperatives [29]. Patronage, which mainly refers to the volume of transactions between members and the cooperative through which the degree of members' contributions can be determined, is crucial in the distribution of surpluses to cooperative members.

The Coase theorem shows that property rights are the true essence of the market. Clear property rights are necessary to realize optimal resource allocation and promote factor input by playing an incentive role [22], which in turn has a positive impact on cooperative business performance. The benefit distribution method of returns according to transactions can enhance members' patronage and encourage them to use the cooperative more frequently to buy and sell agricultural products. Furthermore, the higher the degree and proportion of their use, the greater the surplus members receive, and the higher the income of the cooperative. Simultaneously, it allows members to have the spirit of "mastery" and improves the performance rate and quality of agricultural products, thereby promoting the cooperative's business performance. Furthermore, the return of surplus according to the transaction volume is conducive to establishing a benefit linkage mechanism of "benefit and risk sharing" between cooperatives and their members, which will focus more on cooperatives' operating conditions and strengthen their supervision and management. This will reduce the risk-bearing pressure on cooperative leaders and shares the business risks of the cooperative. Kenkel [30] has opposed the retention of some members' profits as undistributed equity by some agricultural cooperatives, arguing that without the ability to take advantage of tax credits such as DPAD (Domestic Production Activities Deduction), retaining funds as undistributed reserves is the least desirable strategy. Bijman et al. [13] argued that a proportional distribution of earnings generated by internal members and earnings generated from transactions with non-insiders is conducive to the stable growth of the cooperative. Therefore, we propose the following:

Hypothesis 2: The principle of surplus distribution on a trade basis has a catalytic effect on the income of cooperatives compared to cooperatives that do not return on a trade basis.

## 3 Methodology and data

### 3.1 Empirical model

To empirically test the effect of the surplus distribution principle on cooperative development, we construct the following benchmark regression model:

$$SIZE_{it} = \alpha_1 + \alpha_2 SURPLUS_{it} + \sum_{j=3}^{10} \alpha_j C_{it} + \mu_i + \eta_t + \varepsilon_{it}$$

$$INCOME_{it} = \alpha_1 + \alpha_2 SURPLUS_{it} + \sum_{j=3}^{10} \alpha_j C_{it} + \mu_i + \eta_t + \varepsilon_{it}$$

where $i$ denotes province; $t$ denotes year; SIZE denotes the size of cooperative members; INCOME denotes the average income of cooperatives; SURPLUS denotes the surplus distribution principle based on the return of transactions; C denotes a set of control variables that affect cooperative development; $j$ denotes the $j$-th explanatory variable that belongs to the control variables, ranging from 3 to 10; $\mu_i$ denotes the provincial fixed effect; $\eta_t$ denotes the time fixed effect that does not vary over time with individuals; and $\varepsilon_{it}$ is the residual term. α denotes coefficients.

### 3.2 Data

This study uses panel data from 30 provinces across China from 2008 to 2021. Tibet, Hong Kong, Macau, and Taiwan are excluded from the sample owing to data availability. We selected 2008 as the starting year because the Cooperative Law was formally implemented in July 2007, and the number of cooperatives has since increased, whereas data on cooperatives before that date are scarce. Data were obtained from the China Rural Business Management Statistical Yearbook, China Statistical Yearbook, and China Rural Statistical Yearbook.

### 3.3 Variables and descriptive analysis

**3.3.1 Dependent variables.** Membership size (SIZE) captures the organizational size of a cooperative and is measured by dividing the total number of cooperative members in each province by the total number of cooperatives. The average income (INCOME) of a cooperative reflects its performance and is measured by dividing the total income of a cooperative by the total number of cooperatives.

**3.3.2 Independent variables.** According to the provisions of the Cooperative Law of China, we select the proportion of cooperatives that return more than 60% of their distributable surplus (SURPLUS) based on volume as the measure.

**3.3.3 Control variables.** Comprehensively examining the operational status of cooperatives in each province and the influence on their development requires controlling for the effects of other factors. We control for the sales rate of agricultural products (SALE), supply rate of agricultural materials (SUPPLY), degree of branding (BRAND), and rate of firm formation (FIRM). Furthermore, considering the impact of government support on cooperative development, we select the financial support rate (FINANCE) as another control variable. Considering the heterogeneity of agricultural production levels in different provinces, we control for arable land per capita (LAND), agricultural population (AGRPOP), and share of agricultural gross domestic product (AGRPRO) [31]. Table 1 lists the variable definitions and measurements, and Table 2 presents the correlation coefficient matrix.

Table 3 shows serious imbalances in the development level and surplus distribution of cooperatives across provinces. A clear gap exists between the maximum and minimum values of membership size and average income, indicating large

**Table 1. Definitions and measurements of variables.**

| Variables | Definition |
|---|---|
| SIZE | Total number of cooperative members/Total number of cooperatives (persons) |
| INCOME | Total income from cooperatives/total number of cooperatives (10,000RMB) |
| SURPLUS | Number of cooperatives with more than 60% of the distributable surplus returned on a transaction basis/total number of cooperatives (%) |
| SALE | Number of cooperatives selling more than 80% of their agricultural products in a unified manner/total number of cooperatives (%) |
| SUPPLY | Number of cooperatives purchasing more than 80% of their agricultural materials in a unified manner/total number of cooperatives (%) |
| BRAND | Number of cooperatives with registered trademarks/total number of cooperatives (%) |
| FIRM | Number of cooperatives with processing entities/total number of cooperatives (%) |
| FINANCE | Number of cooperatives receiving financial support/total number of cooperatives (%) |
| LAND | Total cultivated area/total population (Mu) |
| AGRPOP | Number of persons engaged in agricultural activities (10,000) |
| AGRPRO | Share of primary sector output in total output (%) |

**Table 2. Correlation test of variables.**

| | SIZE | INCOME | SURPLUS | SALE | SUPPLY | BRAND | FIRM | FINANCE | LAND | AGRPOP | AGRPRO |
|---|---|---|---|---|---|---|---|---|---|---|---|
| SIZE | 1 | | | | | | | | | | |
| INCOME | 0.373*** | 1 | | | | | | | | | |
| SURPLUS | 0.345*** | 0.426*** | 1 | | | | | | | | |
| SALE | 0.290*** | 0.241*** | 0.268*** | 1 | | | | | | | |
| SUPPLY | 0.247*** | 0.287*** | 0.341*** | 0.260*** | 1 | | | | | | |
| BRAND | 0.389*** | 0.681*** | 0.336*** | 0.254*** | 0.182*** | 1 | | | | | |
| FIRM | 0.259*** | 0.376*** | 0.458*** | 0.122** | 0.154*** | 0.379*** | 1 | | | | |
| FINANCE | 0.435*** | 0.630*** | 0.381*** | 0.213*** | 0.257*** | 0.591*** | 0.545*** | 1 | | | |
| LAND | -0.150*** | -0.371*** | -0.326*** | -0.224*** | -0.226*** | -0.397*** | -0.189*** | -0.283*** | 1 | | |
| AGRPOP | 0.163*** | -0.072 | 0.193*** | 0.111** | 0.008 | -0.083* | -0.107** | -0.083* | -0.141*** | 1 | |
| AGRPRO | 0.056 | -0.314*** | -0.286*** | -0.0440 | -0.178*** | -0.198*** | -0.109** | -0.039 | 0.348*** | 0.260*** | 1 |

Notes: ***, **, and * denote significance at 1%, 5%, and 10% levels, respectively.

differences in the development levels of cooperatives in different provinces. This may be caused by the substantial gap in cooperative construction between provinces. In addition, the gap between the maximum and minimum values of surplus distribution is clear, indicating the existence of various distribution systems among different cooperatives across provinces.

## 4 Analysis of the development of farmers' cooperatives

**Fig 2** shows the number and size of cooperatives between 2008 and 2021. The number of cooperatives has shown a continuous growth trend, reaching 2,031,300 by the end of 2021. With the increase in the number of cooperatives, the membership size of cooperatives has shown a decreasing trend, showing a characteristic of "large group, small scale." This is because cooperatives have set low minimum membership thresholds, and inappropriate support policies have created many "shell cooperatives" that seek only to obtain support funds without considering cooperative development and growth.

**Fig 3** shows the total and average income of cooperatives over time. The total income of cooperatives has shown a continuous upward trend, reaching 62.7 billion by the end of 2021. Meanwhile, the average income has shown a

**Table 3. Descriptive statistics of the sample.**

| Variables | Obs. | Mean | Std. dev. | Min | Max |
|---|---|---|---|---|---|
| SIZE | 420 | 53.230 | 45.500 | 7.699 | 339.200 |
| INCOME | 420 | 60.190 | 66.830 | 3.110 | 458.700 |
| SURPLUS | 420 | 15.660 | 8.429 | 1.470 | 65.020 |
| SALE | 420 | 31.400 | 26.570 | 6.442 | 442.200 |
| SUPPLY | 420 | 22.160 | 20.190 | 0.033 | 297.200 |
| BRAND | 420 | 7.162 | 4.502 | 1.705 | 40.950 |
| FIRM | 420 | 3.721 | 3.404 | 0.179 | 25.380 |
| FINANCE | 420 | 5.882 | 6.600 | 0.142 | 48.690 |
| LAND | 420 | 1.277 | 1.023 | 0.104 | 5.031 |
| AGRPOP | 420 | 724.200 | 523.600 | 7.600 | 1981.000 |
| AGRPRO | 420 | 10.070 | 5.359 | 0.231 | 29.990 |

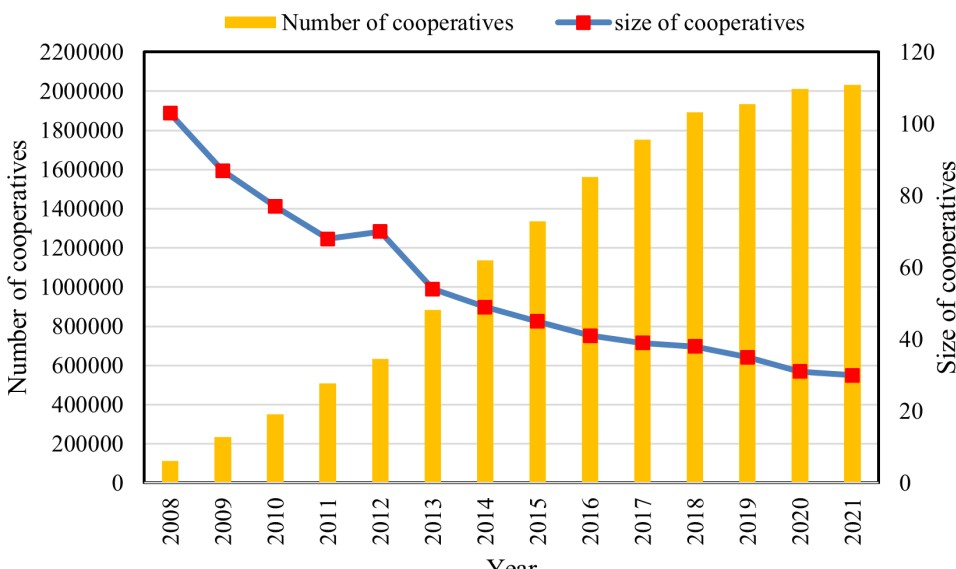

**Fig 2. Number and size of cooperatives.**

decreasing trend. In 2009, the average income showed a particularly large fluctuation because, while the total income of cooperatives experienced a small change that year, the number of cooperatives doubled, preventing income growth from keeping pace with the increase in number and affecting the average income. However, since 2018, total income growth has gradually balanced the growth in membership, mitigating and gradually stabilizing the downward trend in average income. This indicates that cooperatives have moved from high-speed to high-quality development.

**Fig 4** shows that the number of cooperatives with more than 60% of the distributable surplus returned on a trade basis shows a growth trend similar to that of the overall number of cooperatives. By the end of 2021, the number of cooperatives with distributable surplus returned as required reached 332,400. However, the share of these cooperatives with mandatory surplus returns in the total number of cooperatives is low and shows a fluctuating trend, reaching only 16.37% by the end of 2021. This indicates that most cooperatives do not return surpluses in accordance with the relevant provisions of the Cooperative Law.

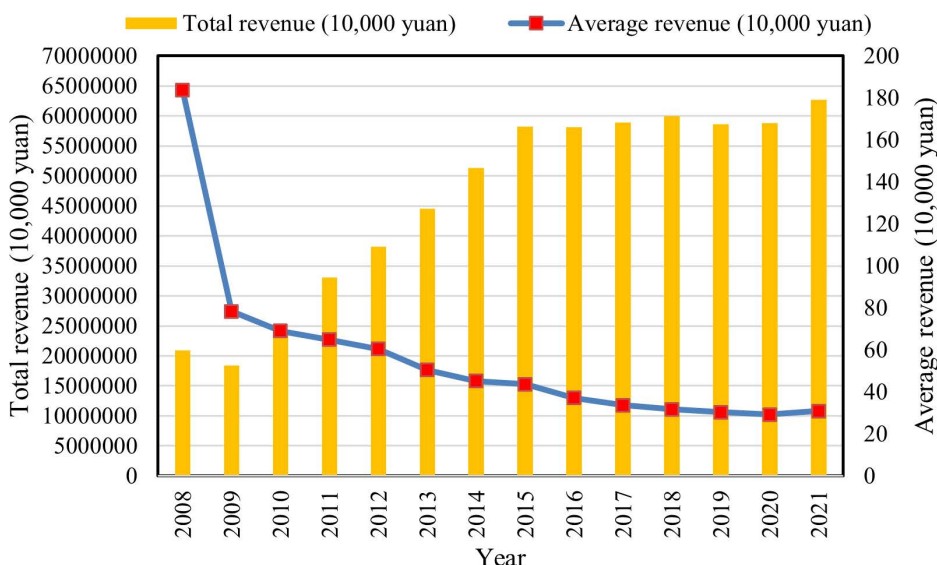

**Fig 3. Total and average cooperative revenue.**

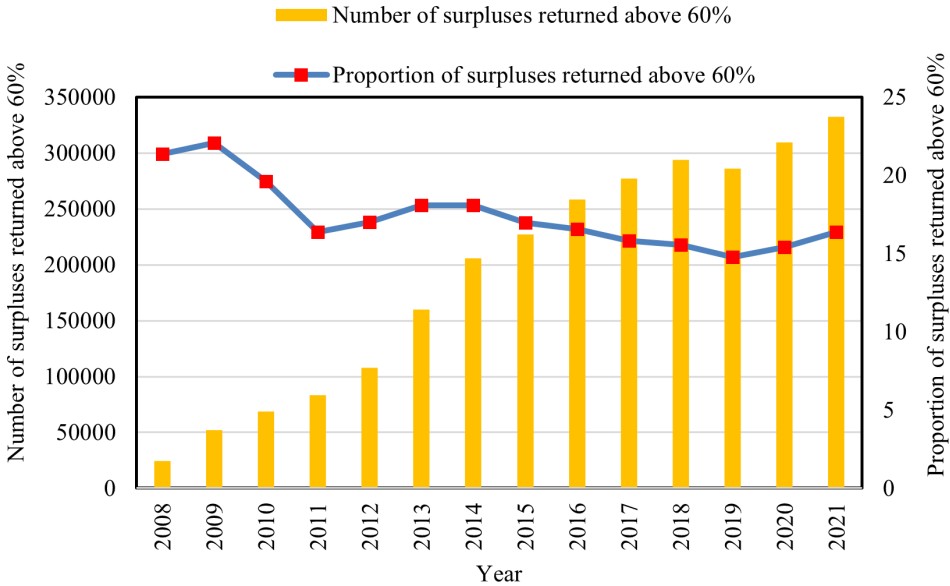

**Fig 4. Number and proportion of surpluses returned above 60%.**

## 5 Impact of the surplus distribution principle on cooperative development

### 5.1 Baseline regression results

The F-test results are significant at the 1% level, strongly rejecting the initial hypothesis and indicating that the model has both time and individual effects (i.e., the choice of a bidirectional fixed effects model). We perform the empirical analysis using Stata 17.0. For a better comparative analysis, columns (1) and (3) of the model show only the regression results of

the core explanatory variables, and a set of control variables is added to the regression in columns (2) and (4) to control for other factors affecting cooperative development.

Columns (1) and (2) of Table 4 show that regardless of whether control variables are added, the impact of the surplus distribution method of return by transaction on the size of cooperatives is significantly correlated. The impact coefficient is negative, and the significance of the regression coefficient is -1.365. Thus, Hypothesis 1 is supported. Columns (3) and (4) show the effects of the normalization of surplus distribution on the average income of cooperatives. The regression results are significant and positively correlated, with a regression coefficient of 2.022, indicating that as the percentage of cooperatives that return on trade based on more than 60% of the distributable surplus increases, the average income of cooperatives increases. Thus, Hypothesis 2 is also supported. These findings show that the principle of surplus distribution based on return on trade has a significant negative effect on the number of cooperative members but a significant positive effect on average cooperative income.

Table 4 shows the regression results with the control variables added. The proportion of agricultural products sold, rate of agricultural supply, degree of branding, and financial support can significantly promote cooperative size growth. Furthermore, the degree of branding can promote cooperative income growth. The effects of the founding rate of the cooperative unit, per capita cultivated area, agricultural population, and share of agricultural production value on the size and income of the cooperative are not significant.

**Table 4. Benchmark regression results.**

| Variables | SIZE | | INCOME | |
| --- | --- | --- | --- | --- |
| | (1) | (2) | (3) | (4) |
| SURPLUS | -0.695*** | -1.365*** | 2.365*** | 2.022*** |
| | (-2.613) | (-4.901) | (5.407) | (4.282) |
| SALE | | 0.077* | | 0.051 |
| | | (1.754) | | (0.692) |
| SUPPLY | | 0.125** | | 0.148 |
| | | (2.143) | | (1.494) |
| BRAND | | 2.908*** | | 4.514*** |
| | | (7.288) | | (6.669) |
| FIRM | | 0.362 | | -1.221 |
| | | (0.693) | | (-1.378) |
| FINANCE | | 1.054*** | | 0.646 |
| | | (3.095) | | (1.124) |
| LAND | | -7.976 | | 10.598 |
| | | (-1.374) | | (1.071) |
| AGRPOP | | -0.044 | | 0.040 |
| | | (-1.309) | | (0.712) |
| AGRPRO | | 1.112 | | 2.112 |
| | | (1.343) | | (1.512) |
| _cons | 64.124*** | 71.570*** | 23.164*** | -71.949* |
| | (14.873) | (2.886) | (3.274) | (-1.712) |
| Province | YES | YES | YES | YES |
| Year | YES | YES | YES | YES |
| N | 420 | 420 | 420 | 420 |
| R² | 0.777 | 0.826 | 0.722 | 0.768 |

Notes: ***, **, and * denote significance at the 1%, 5%, and 10% levels, respectively; robust standard errors are in parentheses.

**5.1.1 Controlling for the estimation method.** To exclude the influence of the estimation method on the regression results, we use (2) and (4) in Table 4 as the basic regression model and adopt the random effect model (RE) and random maximum likelihood estimator (MLE) to conduct the regression. The results in Table 5 show that the estimated coefficients of the effect of the surplus distribution principle on the size of cooperatives are negative, the estimated coefficients of the effect on average income are positive, and both are significant at the 1% level. Compared with the benchmark regression results, only the coefficient size differs slightly, and the choice of estimation method does not change the direction of the sign of the estimated coefficients of surplus distribution and significance. Thus, the regression results are robust.

**5.1.2 Replacing the explanatory variables.** In the benchmark regression, the explanatory variable is replaced by the proportion of cooperatives that return by transaction volume, and the results are shown in columns (1) and (2) of Table 6. With the substitution of the explanatory variable, the estimated coefficients of the standardization of the surplus distribution on the number of cooperative members and average income are significant at the 1% level, and the sign does not change. Thus, the benchmark regression results are relatively robust.

**5.1.3 Shrinking tail treatment.** To eliminate the influence of extreme values, we apply the shrinking tail treatment method by first shrinking all variables at the upper and lower 1% levels and then re-running the fixed effects regression using these values. Columns (3) and (4) of Table 6 show that the signs of the positive and negative effects of the surplus

**Table 5. Results of the robustness test controlling for the estimation method.**

| Variables | RE | | MLE | |
|---|---|---|---|---|
| | SIZE | INCOME | SIZE | INCOME |
| | (1) | (2) | (3) | (4) |
| SURPLUS | -1.274*** | 1.307*** | -1.358*** | 1.293*** |
| | (-4.621) | (3.082) | (-5.002) | (3.064) |
| SALE | 0.099** | 0.075 | 0.097** | 0.075 |
| | (2.157) | (0.979) | (2.193) | (0.982) |
| SUPPLY | 0.142** | 0.188* | 0.139** | 0.188* |
| | (2.322) | (1.831) | (2.348) | (1.859) |
| BRAND | 3.061*** | 5.044*** | 3.024*** | 5.068*** |
| | (7.441) | (7.612) | (7.557) | (7.652) |
| FIRM | 0.018 | -1.052 | 0.105 | -1.055 |
| | (0.043) | (-1.332) | (0.221) | (-1.348) |
| FINANCE | 2.121*** | 3.265*** | 2.101*** | 3.280*** |
| | (7.529) | (7.228) | (7.647) | (7.282) |
| LAND | -16.586*** | -4.272 | -20.062*** | -4.118 |
| | (-4.087) | (-0.913) | (-4.401) | (-0.903) |
| AGRPOP | 0.014 | -0.007 | 0.013 | -0.007 |
| | (1.213) | (-0.727) | (0.929) | (-0.708) |
| AGRPRO | 2.211*** | -0.306 | 2.383*** | -0.409 |
| | (3.264) | (-0.358) | (3.414) | (-0.432) |
| _cons | 21.539* | -4.424 | 26.507* | -3.876 |
| | (1.665) | (-0.315) | (1.756) | (-0.281) |
| Province | YES | YES | NO | NO |
| Year | NO | NO | NO | NO |
| N | 420 | 420 | 420 | 420 |
| R² | 0.161 | 0.581 | – | – |

Notes: ***, **, and * denote significance at the 1%, 5%, and 10% levels, respectively; robust standard errors are in parentheses.

**Table 6. Results of robustness tests replacing the variables and tailoring treatment.**

| Variables | Replacing explanatory variables | | Tailoring treatment | |
|---|---|---|---|---|
| | SIZE | INCOME | SIZE | INCOME |
| | (1) | (2) | (3) | (4) |
| SRUPLUS2 | -0.859*** | 3.106*** | | |
| | (-4.312) | (10.231) | | |
| SURPLUS | | | -1.400*** | 2.294*** |
| | | | (-5.253) | (5.224) |
| SALE | 0.077* | 0.005 | 0.289*** | 0.264* |
| | (1.741) | (0.077) | (3.258) | (1.803) |
| SUPPLY | 0.112* | 0.084 | 0.268*** | 0.238 |
| | (1.904) | (0.942) | (2.636) | (1.417) |
| BRAND | 3.080*** | 3.871*** | 2.572*** | 5.517*** |
| | (7.628) | (6.302) | (6.348) | (8.258) |
| FIRM | -0.110 | -1.014 | 0.646 | -0.682 |
| | (-0.222) | (-1.313) | (1.314) | (-0.841) |
| FINANCE | 0.892*** | 0.139 | 0.602* | 0.845 |
| | (2.658) | (0.269) | (1.848) | (1.577) |
| LAND | -8.548 | 12.604 | -9.959* | 8.963 |
| | (-1.462) | (1.408) | (-1.931) | (1.048) |
| AGRPOP | -0.034 | 0.024 | -0.033 | -0.021 |
| | (-1.022) | (0.469) | (-1.118) | (-0.432) |
| AGRPRO | 1.324 | 2.353* | 1.861** | 2.750** |
| | (1.601) | (1.858) | (2.548) | (2.282) |
| _cons | 62.936** | -92.697** | 53.480** | -55.033 |
| | (2.552) | (-2.471) | (2.472) | (-1.548) |
| Province | YES | YES | YES | YES |
| Year | YES | YES | YES | YES |
| N | 420 | 420 | 420 | 420 |
| R² | 0.856 | 0.818 | 0.856 | 0.818 |

Notes: ***, **, and * denote significance at 1%, 5%, and 10% levels, respectively; robust standard errors are in parentheses.

distribution principle on cooperative membership and average income do not change and remain significant at the 1% level. Therefore, the reduced-tail treatment effectively eliminates the impact of extreme values, further supporting the robustness and reliability of the baseline results.

## 5.3 Endogeneity test

In the baseline regression, we control for the variables that affect cooperative development as comprehensively as possible. However, potential endogeneity issues still require further investigation and primarily stem from two factors. First, reverse causality might lead to estimation bias, in which cooperative membership scale and income growth could conversely drive adjustments in profit distribution policies. Second, unobserved provincial-level heterogeneity (e.g., regional cultural differences) might influence the explanatory and explained variables simultaneously. Thus, this study employs an instrumental variable (IV) approach to mitigate the influence of endogeneity on the estimation results.

We select the historical surplus from the previous year as the IV. First, the IV satisfies the relevance condition with core explanatory variables. A cooperative's historical surplus level directly influences management decisions on distribution

rules, with cooperatives demonstrating strong past surpluses being more likely to maintain high-proportion transaction volume return policies. Table 7 shows the significant explanatory power of IV in the first-stage regression (F-statistic = 28.98), thus rejecting the weak instrument hypothesis. Second, the IV meets the exclusion restriction. The previous year's surplus primarily affects cooperative development through its impact on current distribution policy selection, with a low likelihood of directly influencing the membership scale or income through alternative mechanisms. Specifically, the economic effects of the historical surplus are absorbed by model-controlled fixed assets and regional and time fixed effects, whereas reputation effects are constrained by short-term fluctuations and captured by brand variables.

The two-stage least squares regression results in Table 7 demonstrate that after controlling for endogeneity, the sign and significance level of the estimated coefficients of the core explanatory variables remain robust. Notably, the absolute coefficient values of the core variables increase compared with the baseline regression, suggesting that the ordinary least squares estimates might suffer from downward bias owing to measurement errors or omitted variables. The IV approach not only confirms the robustness of the baseline conclusions but also reveals that conventional regression may underestimate the actual impact of surplus distribution principles.

**Table 7. Endogeneity test results.**

| Variables | First-stage | Second-stage | |
|---|---|---|---|
| | SURPLUS | SIZE | REVENUE |
| IV | 2.024*** | | |
| | (0.52) | | |
| SURPLUS | | -9.410*** | 7.455*** |
| | | (-4.18) | (2.30) |
| SALE | 0.036** | 0.578*** | 0.078 |
| | (0.02) | (3.09) | (0.43) |
| SUPPLY | 0.088*** | 1.036*** | -0.257 |
| | (0.02) | (3.50) | (-0.83) |
| BRAND | 0.194* | 4.310*** | 4.397*** |
| | (0.12) | (4.12) | (4.05) |
| ENTITY | 0.402*** | 4.014*** | -2.852* |
| | (0.10) | (3.16) | (-1.92) |
| FINANCE | 0.140 | 1.909** | 0.002 |
| | (0.09) | (2.50) | (0.01) |
| FARMLAND | 0.444 | -4.828 | 5.657 |
| | (0.97) | (-0.52) | (0.56) |
| AGRPOP | -0.001 | -0.049 | -0.010 |
| | (-0.01) | (-0.88) | (-0.13) |
| AGRPRO | -0.359** | -0.557 | 4.307*** |
| | (-0.15) | (-0.45) | (3.10) |
| _cons | 16.285*** | 29.553 | 71.606** |
| | (-4.39) | (1.17) | (2.28) |
| FIXED UTILITY | YES | YES | YES |
| N | 420 | 420 | 420 |
| R² | 0.867 | 0.800 | 0.783 |
| F | 28.98 | 13.09 | 19.13 |

Notes: ***, **, and * denote significance at the 1%, 5%, and 10% levels, respectively; z-values of the statistics are in parentheses.

## 6 Discussion and policy implications

### 6.1 Discussion

This study examines the theoretical mechanism of the influence of the principle of surplus distribution on cooperative development and draws a map of the development trends of Chinese cooperatives. Accordingly, the impact of the principle of surplus distribution on the development of cooperatives is examined using data on cooperatives in 30 provinces and cities in China from 2008 to 2021. Referring to existing studies, the number and average income of members are chosen as measures of the development level of cooperatives, and the proportion of cooperatives that return more than 60% of their distributable surplus on a transaction-by-transaction basis is selected as a measure of the surplus distribution principle. The results show that the surplus distribution method of transaction-by-transaction returns limits membership growth and promotes increases in cooperative income. These empirical results are shown to be robustness.

Our findings provide three insights into the development of cooperative organizations. First, we provide an evolutionary map of the development of Chinese cooperatives and analyze the trends in and reasons for the development of cooperatives in terms of their number, size, income, and surplus distribution. The results show that agricultural cooperatives in China are characterized by a large number of small cooperatives. The total income of cooperatives has been increasing annually; however, the average income has declined and gradually leveled off. The number of cooperatives trading back more than 60% of their distributable surplus has been increasing, yet the overall proportion remains low. Our findings also provide lessons for cooperative development in other developing and agricultural countries. Second, we use provincial panel data from 30 provinces in China to extend the macro-empirical analysis between the surplus distribution system and cooperative development. Finally, we explain why the strict per-transaction return distribution principle has a significant negative effect on cooperative membership size and a significant positive effect on cooperative income.

The biggest difference between Chinese farmers' cooperatives and those in other developing countries is the heterogeneity in the cooperative membership system. To harmonize the interests of capital and labor, China's Law on Farmers' Specialized Cooperatives allows capital to share a portion of the surplus, but not more than 40%, which takes care of capital interests while allowing laborers to hold the majority of the surplus claim. Our findings also provide important insights into the development of agricultural cooperatives in developing countries that are in transition.

### 6.2 Policy implications

Several policy insights can be drawn based on the findings of this study. First, it is necessary to follow the principle of surplus distribution and strengthen its standardization of surplus distribution. In actual management, we recommend strengthening the training of cooperative leaders, explaining the importance of the surplus distribution system of return according to the volume (amount) of transactions for cooperative development, and encouraging cooperatives to return transactions according to the proportion prescribed by the Cooperative Law.

Second, a property rights system should be built, with producer members as the main body. Regardless of the form of capital, labor, and technology, cooperatives should establish and improve members' account management systems in accordance with the law in specific practice; build a set of rights, responsibilities, and benefits for the property rights system to match and reciprocate intrinsic incentives; and protect the economic interests of members.

Third, it is important to strengthen the linkage of interests between cooperatives and their members and between members and their members to ensure that what members pay for matches what they receive. Cooperatives should guide and encourage their members to establish a model of "shared benefits and risks" and make the return of patronage the main surplus distribution method to reduce the number of "free-riders".

## 7 Limitations and future recommendations

Although this study provides some encouraging conclusions and findings, it also has limitations that offer potential for future research. First, the impact of surplus distribution principles on cooperative development is interesting. However, we

only focus on the dimension of surplus distribution and do not include democratic management, education and training, or risk and value perception. For example, as the duration of education and training increases, does the productive capacity and willingness of community members to specialize increase as their professional capital accumulates [32]? Owing to a lack of data, we cannot currently include these supports in our model. However, more comprehensive data may become available as database construction develops, and further conclusions could then be drawn from examining each cooperative development principle.

Second, we must explore the impact of surplus distribution principles on cooperative development at the micro level. The data in this study are from panel data of each province; however, a micro-level investigation is lacking. In the future, cooperatives can be categorized in more detail, and general patterns can be analyzed by conducting research on several different types of cooperatives. Furthermore, whether the principle of surplus distribution yields different results for the development of different types of cooperatives can be explored.

## Supporting information

**S1 Data. Experimental measurements and source data for figures in Impact of the surplus distribution principle on the development of agricultural cooperatives in China.**
(XLSX)

## Author contributions

**Conceptualization:** Yating Nie, Aihua Lv, Lei Ye.

**Data curation:** Yating Nie, Aihua Lv, Lei Ye.

**Formal analysis:** Yating Nie, Aihua Lv, Lei Ye.

**Funding acquisition:** Yating Nie, Aihua Lv, Lei Ye.

**Investigation:** Yating Nie, Aihua Lv, Lei Ye.

**Methodology:** Yating Nie, Aihua Lv, Lei Ye.

**Project administration:** Yugang HAN.

**Resources:** Yugang HAN.

**Software:** Yugang HAN.

**Supervision:** Yugang HAN.

**Validation:** Yugang HAN.

**Visualization:** Yugang HAN.

**Writing – original draft:** Yugang HAN.

**Writing – review & editing:** Yugang HAN.

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
