## [Decision Letter · Decision Letter 0]

1 Dec 2024

PONE-D-24-44710The Impact of the Surplus Distribution Principle on the Development of Agricultural Cooperatives in ChinaPLOS ONE

Dear Dr. HAN,

Thank you for submitting your manuscript to PLOS ONE. After careful consideration, we feel that it has merit but does not fully meet PLOS ONE’s publication criteria as it currently stands. Therefore, we invite you to submit a revised version of the manuscript that addresses the points raised during the review.

One of the referees recommends only minor revisions and is generally satisfied with the paper. However, the second referee suggests major revisions and raises concerns, that you need to properly address for the revision to eventually be considered for publication.

We look forward to receiving your revised manuscript.

Kind regards,

Marco Maria Sorge, PhD

Academic Editor

PLOS ONE

3. Thank you for stating the following financial disclosure: [Funded by: Anhui Normal University Talent Cultivation Fund Project "Research on the Activation of Rural

Collective Economy Development by New Cooperatives Led by Party Organizations" (QZJDBN2021XZC04);

Anhui Provincial Higher Education Research Project (Major Project) "Research on the Mechanisms and Effects of

Collaborative Development between Anhui's Wanjiang and Northern Anhui Regions—Based on a

'Subject-Industry-Space' Collaborative Perspective" (2023AH040017).]. Please state what role the funders took in the study. If the funders had no role, please state: "The funders had no role in study design, data collection and analysis, decision to publish, or preparation of the manuscript." If this statement is not correct you must amend it as needed. Please include this amended Role of Funder statement in your cover letter; we will change the online submission form on your behalf.

5. We note that you have indicated that there are restrictions to data sharing for this study. For studies involving human research participant data or other sensitive data, we encourage authors to share de-identified or anonymized data. However, when data cannot be publicly shared for ethical reasons, we allow authors to make their data sets available upon request. For information on unacceptable data access restrictions, please see http://journals.plos.org/plosone/s/data-availability#loc-unacceptable-data-access-restrictions. Before we proceed with your manuscript, please address the following prompts: a) If there are ethical or legal restrictions on sharing a de-identified data set, please explain them in detail (e.g., data contain potentially identifying or sensitive patient information, data are owned by a third-party organization, etc.) and who has imposed them (e.g., a Research Ethics Committee or Institutional Review Board, etc.). Please also provide contact information for a data access committee, ethics committee, or other institutional body to which data requests may be sent. b) If there are no restrictions, please upload the minimal anonymized data set necessary to replicate your study findings to a stable, public repository and provide us with the relevant URLs, DOIs, or accession numbers. Please see http://www.bmj.com/content/340/bmj.c181.long for guidelines on how to de-identify and prepare clinical data for publication. For a list of recommended repositories, please see https://journals.plos.org/plosone/s/recommended-repositories. You also have the option of uploading the data as Supporting Information files, but we would recommend depositing data directly to a data repository if possible. Please update your Data Availability statement in the submission form accordingly.

6. PLOS requires an ORCID iD for the corresponding author in Editorial Manager on papers submitted after December 6th, 2016. Please ensure that you have an ORCID iD and that it is validated in Editorial Manager. To do this, go to ‘Update my Information’ (in the upper left-hand corner of the main menu), and click on the Fetch/Validate link next to the ORCID field. This will take you to the ORCID site and allow you to create a new iD or authenticate a pre-existing iD in Editorial Manager.

Additional Editor Comments (if provided):

Reviewers' comments:

Reviewer's Responses to Questions

**Comments to the Author**

1. Is the manuscript technically sound, and do the data support the conclusions?

Reviewer #1: Yes

Reviewer #2: Yes

2. Has the statistical analysis been performed appropriately and rigorously? 

Reviewer #1: Yes

Reviewer #2: Yes

3. Have the authors made all data underlying the findings in their manuscript fully available?

Reviewer #1: Yes

Reviewer #2: Yes

4. Is the manuscript presented in an intelligible fashion and written in standard English?

Reviewer #1: Yes

Reviewer #2: Yes

5. Review Comments to the Author

Reviewer #1: Reviewer 1 : The abstract should answer why your research is necessary? Why are these findings useful and important? The current style of writing has little impact. Although the author emphasizes the important influence of earnings distribution, we cannot get a preliminary definition from the abstract. Is earnings distribution the surplus of unified operation or the surplus of product profit difference? In addition, the summary does not indicate the size of the survey sample.

Reviewer 2 : This Cooperative surplus distribution problems in India are indeed worthy of study, and could be raised to the level of the developing countries' presentation in the introduction to enhance the generality of the article's conclusions.

Reviewer 3: The literature citations are very old, and some literatures before 2015 do not belong to the important authoritative literatures and should be replaced. may refer: https://doi.org/10.3390/ijerph19106255. https://doi.org/10.1007/s10668-022-02744-2. https://doi.org/10.3390/agriculture12081145. https://doi.org/10.1016/j.jclepro.2022.135068.

Reviewer 4� In line 80, the author argues that no one has studied the effect of surplus distribution on the scale of cooperation and membership dynamics. Generally speaking, no one's research direction may lack significance. The author should elaborate more on why this topic research is meaningful and what reference suggestions it provides for policy making.

Reviewer 5: In the theoretical analysis, the author introduced the concept of property rights system, and spent a lot of space from the company to the cooperative society, which is very good. However, when it comes to the cooperative property rights surplus distribution system, there is a lack of corresponding literature support, and the elaboration is less.

Reviewer 6: 160 lines of discussion on the size of cooperative members, I suggest that we can draw an optimal size graph, which will be more vivid.

Reviewer 7: The formulation of hypothesis 1 is too cumbersome and should be condensed

Reviewer 8: The endogeneity test part can be advanced.

Reviewer 9: The author needs to design and write a “Discussion” section, where the author shows how the work advances the field from its current state of knowledge. The similarities, differences and innovative findings of this paper with previous studies. I suggest the authors develop a new section called "Limitations and Future recommendations". Study limitations are the constraints placed on the ability to generalize from the results, to describe applications further to practice.

Reviewer #2: Review of manuscript PONE-D-24-44710

The Impact of the Surplus Distribution Principle on the Development of Agricultural Cooperatives in China

Summary.

This article discusses the impact of the surplus distribution principle on the development of farmers' cooperatives, with the aim of evaluating the effectiveness of the surplus distribution method based on the return of transactions and proposing policy recommendations for the development of cooperatives. Based on this, the data of cooperatives from 2008-2021 are used to depict the development pattern and surplus distribution status of cooperatives, and the two-way fixed utility model is used as a benchmark to study the impact of the surplus distribution principle on the development of cooperatives.

Comments

1. Introduction and Research Hypothesis:

According to what I've read in the paper, you wish to assess how surplus affects the "development of farmers' cooperatives." The literature you cited regarding the significance of surplus is very broad and sections 1 and 2.1 provide an overview of it.

I have some concerns about several aspects.

2. Methodology and Data

1) I suggest to better explain the computation of your main independent variables: “surplus" defined as “number of cooperatives with more than 60% of distributable surplus on a transaction basis/total number of cooperatives (%)” - see Table 1.

2) I also have a concern about the dependent variable since I believe it could be more appropriate considerer other dependent variables in the equation. “Size "(total number of cooperative members/total number of cooperatives (persons)) is obviously a key variable in the analysis but the Number of Cooperatives is included among the different independent variables” sales; delivery; brand; business; finance" as the denominator. Because of this, I don't think that this represents a good indicator.

3) Revenue (Total income from cooperatives/total number of cooperatives (10,000RMB)) is a proxy of development, but it could be that costs are very high and therefore this affects development and performance and consequently welfare.

I am not familiar with your dataset and the information it contains, but it might be interesting to add other proxies of cooperatives development, depending on the availability of data, for example:

•Sales Growth rate (you included “the sales rate of agricultural products (sales)” as control)

•Sales/N Members (you included “the sales rate of agricultural products (sales)” as control)

•Profit

•Profit/N Members

Or -If available:

•Average Value Added

Of course, considering the lagged independent variables

4) In addition, given the high correlation between the variables, I suggest adding the correlation matrix in the text.

3. Econometric approach

1) My concern is related to Section 5.3 about the endogeneity. As I read “In this paper, the explanatory variables are lagged by one period as instrumental variables, and regression analyses are conducted using two-stage least squares (2SLS) with the reduced-form data”.

I appreciate your efforts to explain the empirical approach, the IV, and other details, but given the issue that you also highlighted, I believe that in this case, an instrument different from lag variable is required to address the endogeneity.

4. Additional points

•Comparison -from a descriptive point of view -with other countries.

•Compute a map that illustrates China's agricultural cooperative distribution system.

•Think about how the geographical area or regions differ from one another.

6. PLOS authors have the option to publish the peer review history of their article (what does this mean? ). If published, this will include your full peer review and any attached files.

**Do you want your identity to be public for this peer review?** For information about this choice, including consent withdrawal, please see our Privacy Policy .

Reviewer #1: No

Reviewer #2: No

---

## [Author Response · Author response to Decision Letter 1]

16 Jan 2025

Dear Editors and Reviewers:

Thank you for giving us the opportunity to submit a revised manuscript entitled "A Study on the Impact of the Principle of Surplus Distribution on the Development of peasant Cooperatives in China" for publication in PLOS ONE.We appreciate the time and effort that you and the reviewers dedicated to providing feedback on our manuscript and are grateful for the insightful comments on and valuable improvements to our paper. All suggestions made by you and the reviewers have been incorporated into the manuscript. Those changes are marked in red within the manuscript. Please see below a point-by-point response to your and the reviewers' comments and concerns. The authors welcome further constructive comments if any.

Response to Associate Editor

# Comment 1. Thank you for your contribution to PLOS ONE. After careful consideration, we believe it has merit, but does not currently fully meet PLOS ONE's criteria for publication. We therefore invite you to submit a revised manuscript to address the issues raised during the review process.One of the reviewers suggested only minor changes and was generally satisfied with the paper. However, the second reviewer suggests significant changes and raises a number of questions that you need to address correctly in order to finally consider publishing.

# Response 1. Thank you very much for your time and consideration. While we did not receive your specific comments about the article, we appreciate the reviewers for their valuable input. We have carefully considered their suggestions and made the necessary changes to address their concerns. If you have any other feedback or suggestions, we would appreciate it.

Response to Reviewer 1

This paper studies the impact of normalized earnings distribution on the development of cooperatives based on the two-way fixed utility model, aiming to evaluate the validity of the surplus distribution method based on transaction income. The results show that the distribution of surplus strictly according to the distributable surplus of more than 60% has a significant negative impact on the scale of members of cooperatives, but has a significant positive impact on cooperative income. This study helps to understand the impact of the principle of surplus distribution on the performance and scale of farmers' cooperatives from the macro level, and provides important policy implications for promoting the standardized construction of farmers' surplus distribution and guiding members to establish a stable interest linkage mechanism with cooperatives.

Here are some comments:

# Comment 1. The abstract should answer why your research is necessary? Why are these findings useful and important? The current style of writing has little impact. Although the author emphasizes the important influence of earnings distribution, we cannot get a preliminary definition from the abstract. Is earnings distribution the surplus of unified operation or the surplus of product profit difference? In addition, the summary does not indicate the size of the survey sample.

# Response 1. Thank you very much for this great comment!After modification, we indicate in the abstract that the survey data of this paper comes from “Provincial Panel Data on Cooperative Development and Surplus Distribution in 30 Provinces in China, 2008-2021”�Line 9–10�. The analysis of the results shows that, theoretically, the surplus distribution method, which is strictly based on the trading return of more than 60% of the distributable surplus, has a significant negative effect on the membership size of cooperatives, while it has a significant positive effect on the income of cooperatives. In terms of practice Line 18–22��“this study helps to understand the impact of surplus distribution principles on the performance and size of farmers' cooperatives at the macro level, and provides important policy insights for promoting the standardization of surplus distribution in farmers' cooperatives and guiding members to establish a stable benefit linkage mechanism with cooperatives”.In the introduction Line 62–68�, we add the definition of distributable surplus from China's Law on Farmers' Specialized Cooperatives.

# Comment 2. This Cooperative surplus distribution problems in India are indeed worthy of study, and could be raised to the level of the developing countries' presentation in the introduction to enhance the generality of the article's conclusions.

# Response 2. Thank you very much for this great comment!In the introduction Line36–38�, we add a comparison of farmers' professional cooperatives in developing and developed countries and provide an overview of the characteristics of farmers' professional cooperatives in developing countries. At the same time, we enriched the article with a description of the development of farmers' specialized cooperatives in developing countries Line 38–43�: “Because of the diverse political, economic, and socio-cultural conditions in each country, the trajectory of cooperative development has been quite different, and the patterns of cooperative development and distribution have shown significant variability”. Following this, we have selected relevant studies of agricultural cooperatives in India, El Salvador, Peru and Viet Nam with regard to the distribution of earnings or surpluses to enhance the generality of the findings of the article Line 44–58�.

# Comment 3. The literature citations are very old�and some literatures before 2015 do not belong to the important authoritative literatures and should be replaced. may refer�https://doi.org/10.3390/ijerph19106255.https://doi.org/10.1007/s10668-022-02744-2.https://doi.org/10.3390/agriculture12081145.https://doi.org/10.1016/j.jclepro.2022.135068.

# Response 3. Thank you very much for this great comment In response to your request, we have removed eleven non-authoritative citations and retained only four relatively important pre-2015 citations. At the same time, we have added fifteen post-2015 cited literatures, including four provided by the reviewers, based on adjustments to the content of the article.

# Comment 4. In line 97-106, the author argues that no one has studied the effect of surplus distribution on the scale of cooperation and membership dynamics. Generally speaking, no one's research direction may lack significance. The author should elaborate more on why this topic research is meaningful and what reference suggestions it provides for policy making.

# Response 4. Thank you very much for this valuable Comment! We have added a discussion of the significance of this study Line 90–93�: “This paper aims to narrow these gaps by investigating the impact of the surplus distribution principle, as stipulated in China’s Cooperative Law, on the development of agricultural cooperatives. Specifically, we focus on two key dimensions: the size and income of cooperatives.” In the third part of the research contribution, we add the implications of this study for policy making:�Line 104–106�: “This study provides important policy insights for promoting the standardization of surplus distribution in developing countries, and for guiding members to establish a stable mechanism for linking interests with cooperatives.”.

# Comment 5. In the theoretical analysis, the author introduced the concept of property rights system, and spent a lot of space from the company to the cooperative society, which is very good. However, when it comes to the cooperative property rights surplus distribution system, there is a lack of corresponding literature support, and the elaboration is less.

# Response 5. Thank you very much for this great comment! First, we add a discourse on the nature of surplus distribution systems and property rights in cooperatives: “The type of distribution system adopted by cooperatives is essentially related to the issue of surplus control and surplus claim rights in cooperatives”�Line 134–136�. Secondly, the issue based on cooperative member ownership is added (Line 189–193): “In a cooperative property rights system, granting ownership to cooperative members may have beneficial incentive effects, but the distribution of benefits and surpluses among members may also lead to free-rider problems within the firm, which can negate the incentive effects of cooperative ownership.” And we add Kenkel and Bijman's views on the distribution of cooperative surpluses (Line 236–242).

# Comment 6. 160 lines of discussion on the size of cooperative members, I suggest that we can draw an optimal size graph, which will be more vivid.

# Response 6. Following the reviewers' comments, we have added a plot of the relationship between cooperative size and performance. This relationship graph is an inverted U-shaped curve (Line 190–190).

# Comment 7. The formulation of hypothesis 1 is too cumbersome and should be condensed

# Response 7. Thank you for pointing this out! In accordance with the reviewer's suggestion, we have simplified hypothesis 1 to read (Line 204–206): “The principle of surplus distribution on a trade basis will have a dampening effect on the size of cooperatives compared to cooperatives that do not return on a trade basis”. In addition, in order to maintain the balance between hypotheses 1 and 2, we simplified hypothesis 2 to read: “The principle of surplus distribution on a trade basis has a catalytic effect on the income of cooperatives compared to cooperatives that do not return on a trade basis”�Line 244–246

# Comment 8. The endogeneity test part can be advanced.

# Response 8. Thank you for your valuable suggestions on our paper, especially your attention to the issue of endogeneity. Based on your feedback, we will further consider using alternative instrumental variables to address the endogeneity problem. Following your suggestion, we will use the cooperative's historical surplus from the previous year as a new instrument to replace the lagged explanatory variable in Section 5.3, in order to effectively resolve the endogeneity issue (Line 410–418): “To avoid serious endogeneity problems that could have biased and inconsistent effects on the estimated coefficients, this paper takes measures to deal with endogeneity. In this paper, the historical surplus of the cooperative in the previous year was used as an instrumental variable and regression analysis was conducted using two-stage least squares (2SLS) method using the reduced-form data. The regression results are presented in Table 7 and show that the sign and significance level of the coefficients of the parameter estimates of cooperative size and income are unchanged by strictly trading back more than 60% of the distributable surplus, indicating that the results of the benchmark regression are robust.” We have also updated the results of the endogeneity test, as shown in Table 7 (Line 419).

# Comment 9. The author needs to design and write a “Discussion” section, where the author shows how the work advances the field from its current state of knowledge. The similarities, differences and innovative findings of this paper with previous studies. I suggest the authors develop a new section called "Limitations and Future recommendations". Study limitations are the constraints placed on the ability to generalize from the results, to describe applications further to practice.

# Response 9. We thank the authors for their interest in the concluding section of the paper! At the suggestion of the authors, the title of section VI has been changed from “Conclusions and Policy Implications” to “Discussion and Policy Implications”. Furthermore, we have divided section VI into two parts: “Discussion” and “Policy Implications”. In the second paragraph of the “Discussion” section (Line 436), we show that the results of the study provide three research perspectives on the development of cooperative organizations. In the fourth paragraph of the “Policy Implications” section, we have added a third policy implication (Line 478-479): “Thirdly, it is important to strengthen the linkage of interests between cooperatives and their members, and between members and their members, to ensure that members get what they pay for.”. On this basis, we have added a section 7 entitled “Limitations and Future Recommendations” (Line 491) in order to summarize the limitations of this study and suggest future directions for research and practice. In the second paragraph of this section, we describe one of the directions for future research with a real-world example (Line 478-479): “For example, as the duration of education and training increases, does the productive capacity and willingness of community members to specialize increase as their professional capital accumulation becomes richer?” Second, in the third paragraph, we also suggest some directions for future research (Line 507–510): “In the future, the types of cooperatives can be categorized in more detail, and by conducting research on a number of cooperatives of different types, general patterns can be analyzed and whether the principle of surplus distribution has different results for the development of different types of cooperatives can be explored.”.

Response to Reviewer 2

# Comment 1. Introduction and Research Hypothesis:According to what I've read in the paper, you wish to assess how surplus affects the "development of farmers' cooperatives." The literature you cited regarding the significance of surplus is very broad and sections 1 and 2.1 provide an overview of it.

# Response 1. Thank you very much for your recognition! However, since you did not put forward any specific modification suggestions for this part, it was only partially modified according to the opinions of the first reviewer.

# Comment 2. Methodology and Data

# Comment 2.1. I suggest to better explain the computation of your main independent variables: “surplus" defined as “number of cooperatives with more than 60% of distributable surplus on a transaction basis/total number of cooperatives (%)” - see Table 1.

# Response 2.1. Thank you for your valuable comments, especially regarding the calculation method of the "surplus" variable. We greatly appreciate your attention to this detail. Regarding the calculation method you mentioned for the "surplus" variable, specifically, this variable represents the proportion of cooperatives in which the surplus allocated exceeds 60% of total cooperatives. We first calculate the surplus allocated by each cooperative in each trading period, then identify how many of these cooperatives allocate more than 60% of their surplus. Finally, we divide the number of cooperatives meeting this criterion by the total number of cooperatives to obtain the percentage of this variable. To ensure that readers can clearly understand the calculation process of this variable, we will further clarify this point in the paper and add the corresponding explanation in Table 1 (Line 306). We sincerely thank you again for your careful review of the paper's details and look forward to your further feedback.

# Comment 2.2. I also have a concern about the dependent variable since I believe it could be more appropriate considerer other dependent variables in the equation. "Size" (total number of cooperative members/total number of cooperatives (persons)) is obviously a key variable in the analysis but the Number of Cooperatives is included among the different independent variables” sales; delivery; brand; business; finance" as the denominator. Because of this, I don't think that this represents a good indicator.

# Response 2.2. Thank you for your detailed review of our paper, particularly your valuable comments regarding the choice of the dependent variable. Regarding your concern about the dependent variable "Size" we believe that, within the current research framework, "Size" (the ratio of total cooperative members to the number of cooperatives) is an appropriate key variable. It reflects the scale of cooperatives and serves as a summary of other critical variables when analyzing the overall performance of cooperatives. We understand your concern that "Number of Cooperatives " is already considered in the denominator of an independent variable. However, we still believe that the use of "Cooperative Size" in our model remains justified, as it independently provides informati

---

## [Decision Letter · Decision Letter 1]

2 Mar 2025

PONE-D-24-44710R1Impact of the Surplus Distribution Principle on the Development of Agricultural Cooperatives in ChinaPLOS ONE

Dear Dr. HAN,

Thank you for submitting your manuscript to PLOS ONE. After careful consideration, we feel that it has merit but does not fully meet PLOS ONE’s publication criteria as it currently stands. Therefore, we invite you to submit a revised version of the manuscript that addresses the points raised during the review process.

I would like you to provide some further discussion about the endogeneity issues involved in the empirical analysis as well as in support of the IV strategy adopted to address those, as suggested by Reviewer #2.

We look forward to receiving your revised manuscript.

Kind regards,

Marco Maria Sorge, PhD

Academic Editor

PLOS ONE

Journal Requirements:

Reviewers' comments:

Reviewer's Responses to Questions

**Comments to the Author**

1. If the authors have adequately addressed your comments raised in a previous round of review and you feel that this manuscript is now acceptable for publication, you may indicate that here to bypass the “Comments to the Author” section, enter your conflict of interest statement in the “Confidential to Editor” section, and submit your "Accept" recommendation.

Reviewer #1: All comments have been addressed

Reviewer #2: All comments have been addressed

2. Is the manuscript technically sound, and do the data support the conclusions?

Reviewer #1: Yes

Reviewer #2: Yes

3. Has the statistical analysis been performed appropriately and rigorously? 

Reviewer #1: Yes

Reviewer #2: Yes

4. Have the authors made all data underlying the findings in their manuscript fully available?

Reviewer #1: Yes

Reviewer #2: (No Response)

5. Is the manuscript presented in an intelligible fashion and written in standard English?

Reviewer #1: Yes

Reviewer #2: Yes

6. Review Comments to the Author

Reviewer #1: (No Response)

Reviewer #2: Thank you for the opportunity to read the manuscript revised version.

I appreciated the authors effort in improve the paper contents and I understand the difficult to obtain additional data.

However, I feel greater attention should be given to the issue of endogeneity presented by the author in section 5.3. In particular, I would recommend expanding the discussion on the use of instrumental variable and the exclusion restriction, providing additional details and argument on the reason for employing the aforementioned instrument. Finally, I suggest showing the first stage findings.

7. PLOS authors have the option to publish the peer review history of their article (what does this mean? ). If published, this will include your full peer review and any attached files.

**Do you want your identity to be public for this peer review?** For information about this choice, including consent withdrawal, please see our Privacy Policy .

Reviewer #1: No

Reviewer #2: No

---

## [Author Response · Author response to Decision Letter 2]

14 Apr 2025

Dear Editors and Reviewers:

Thank you very much for giving us the opportunity to submit a revised draft of the manuscript entitled "Impact of the surplus distribution principle on the development of agricultural cooperatives in China " (ID: HSSCOMMS-19456R1) for publication in PLOS ONE. We appreciate the time and effort that you and the reviewers dedicated to providing feedback on our manuscript. All suggestions made by you and the reviewers have been incorporated into the manuscript. Please see below a point-by-point response to your and the reviewers' comments and concerns.

Response to Associate Editor

# Comment 1. I would like you to provide some further discussion about the endogeneity issues involved in the empirical analysis as well as in support of the IV strategy adopted to address those, as suggested by Reviewer #2.

# Response 1. Thank you very much for your time and consideration. While we haven't received any comments from you, we appreciate the valuable comments provided by the reviewers. We have carefully considered their suggestions and have made the necessary revisions to address their concerns. If you have any additional feedback or suggestions, we would be grateful to receive them.

Response to Reviewer 1

# Comment 1. No Response

# Response 1. We would like to thank you for your valuable comments, which have significantly improved the quality of the manuscript. Please let us know if there are any errors in the revision of the manuscript. The authors will revise it as soon as possible.

Response to Reviewer 2

# Comment 1. Thank you for the opportunity to read the manuscript revised version. I appreciated the authors effort in improve the paper contents and I understand the difficult to obtain additional data. However, I feel greater attention should be given to the issue of endogeneity presented by the author in section 5.3. In particular, I would recommend expanding the discussion on the use of instrumental variable and the exclusion restriction, providing additional details and argument on the reason for employing the aforementioned instrument. Finally, I suggest showing the first stage findings.

# Response 1. Thank you very much for pointing this out. We have expanded the discussion on the use of the instrumental variable and the exclusion restriction as suggested by the reviewer: "We select the historical surplus from the previous year as the IV. First, the IV satisfies the relevance condition with core explanatory variables. A cooperative's historical surplus level directly influences management decisions on distribution rules, with cooperatives demonstrating strong past surpluses being more likely to maintain high-proportion transaction volume return policies. Table 7 shows the significant explanatory power of IV in the first-stage regression (F-statistic = 28.98), thus rejecting the weak instrument hypothesis. Second, the IV meets the exclusion restriction. The previous year's surplus primarily affects cooperative development through its impact on current distribution policy selection, with a low likelihood of directly influencing the membership scale or income through alternative mechanisms. Specifically, the economic effects of the historical surplus are absorbed by model-controlled fixed assets and regional and time fixed effects, whereas reputation effects are constrained by short-term fluctuations and captured by brand variables. The two-stage least squares regression results in Table 7 demonstrate that after controlling for endogeneity, the sign and significance level of the estimated coefficients of the core explanatory variables remain robust. Notably, the absolute coefficient values of the core variables increase compared with the baseline regression, suggesting that the ordinary least squares estimates might suffer from downward bias owing to measurement errors or omitted variables. The IV approach not only confirms the robustness of the baseline conclusions but also reveals that conventional regression may underestimate the actual impact of surplus distribution principles."

Thank you again for your valuable suggestions to improve the quality of our manuscript. If there any other modifications we could make, we would like very much to modify them and we really appreciate your help.

---

## [Editor Report · Decision Letter 2]

16 Apr 2025

Impact of the Surplus Distribution Principle on the Development of Agricultural Cooperatives in China

PONE-D-24-44710R2

Dear Dr. HAN,

We’re pleased to inform you that your manuscript has been judged scientifically suitable for publication and will be formally accepted for publication once it meets all outstanding technical requirements.

Kind regards,

Marco Maria Sorge, PhD

Academic Editor

PLOS ONE
---

## [Editor Report · Acceptance letter]

PONE-D-24-44710R2

PLOS ONE

Dear Dr. HAN,

I'm pleased to inform you that your manuscript has been deemed suitable for publication in PLOS ONE. Congratulations! Your manuscript is now being handed over to our production team.

Kind regards,

on behalf of

Professor Marco Maria Sorge

Academic Editor

PLOS ONE